# Goodness-of-Fit Test for the Bivariate Hermite Distribution

Pablo González-Albornoz [1] and Francisco Novoa-Muñoz [2,*]

1   Departamento de Matemática, Universidad Adventista de Chile, Chillán 3780000, Chile
2   Departamento de Estadística, Universidad del Bío-Bío, Concepción 4051381, Chile
*   Correspondence: fnovoa@ubiobio.cl; Tel.: +56-413111310

**Abstract:** This paper studies the goodness of fit test for the bivariate Hermite distribution. Specifically, we propose and study a Cramér–von Mises-type test based on the empirical probability generation function. The bootstrap can be used to consistently estimate the null distribution of the test statistics. A simulation study investigates the goodness of the bootstrap approach for finite sample sizes.

**Keywords:** bivariate Hermite distribution; goodness-of-fit; empirical probability generating function; bootstrap distribution estimator





## 1. Introduction

Testing the goodness-of-fit (gof) of given observations with a probabilistic model is a crucial aspect of data analysis.

Since the chi-square test was proposed and analyzed by Pearson in 1900 until today, new gof tests have been constructed and applied to continuous and discrete data. Just to mention some of the most recent publications, there are, for example, the works of: Ebner and Henze [1], Górecki, Horváth and Kokoszka [2], Puig and Weiß [3], Arnastauskaitè et al. [4], Dörr, Ebner, and Henze [5]), Kolkiewicz, Rice, and Xie [6], Milonas et al. [7], Di Noia et al. [8], and Erlemann and Lindqvist [9].

Because count data can appear in different circumstances, the present investigation is oriented to gof in the discrete case, specifically, in the bivariate Hermite distribution (BHD).

In the univariate configuration, the Hermite distribution is a linear combination of the form $Y = X_1 + 2X_2$, where $X_1$ and $X_2$ are independent Poisson random variables. The distinguishing property of the univariate Hermite distribution (UHD) is that it is flexible when it comes to modeling count data that present a multimodality, in addition to presenting several zeros, which is called zero-inflation. It also allows for modeling data in which the overdispersion is moderate, that is, the variance is greater than the expected value. It was McKendrick at [10] who modeled a phagocytic experiment (bacteria count in leukocytes) through the UHD, obtaining a more satisfactory model than with the Poisson distribution. However, in practice, bivariate count data emerge in several different disciplines and the BHD plays an important role, having superinflated data—for example, the number of accidents in two different periods [11].

The only gof test related to the Hermite distribution found in this study so far is the one developed by the researchers Meintanis and Bassiakos in [12]. However, this test is for univariate data.

On the other hand, to the best of our knowledge, we did not find literature on gof tests for BHD.

The purpose of this paper is to propose and study a gof test for the bivariate Hermite Distribution that is consistent.

According to Novoa-Muñoz in [13], the probability generating function (pgf) characterizes the distribution of a random vector and can be estimated consistently by the empirical probability generating function (epgf); the proposed test is a function of the epgf. This statistical test compares the epgf of the data with an estimator of the pgf of the BHD.

As it is well known, to establish the rejection region, we need to know the distribution of the statistic test.

As for finite sample sizes, the resulting test statistic is of the Cramér–von Mises type, and it was not possible to calculate explicitly the distribution of the statistic under a null hypothesis. This is why one uses simulation techniques. Therefore, we decided to use a null approximation of the statistic by using a parametric bootstrap.

Because the properties of the proposed test are asymptotic (see, for example, [14]) and with the purpose of evaluating the behavior of the test for samples of finite size, a simulation study was carried out.

The present work is ordered as follows: In Section 2, we present some preliminary results that will serve us in the following chapters, and the definition of the BHD with some of its properties is also given. In Section 3, the proposed statistic is presented. Section 4 is devoted to showing the bootstrap estimator and its approximation to the null distribution of the statistic. Section 5 is dedicated to presenting the results of a simulation study, power of a hypothesis test, and the application to a set of real data.

Before ending this section, we introduce some notation: $\mathcal{F}_A \underset{\delta}{\wedge} \mathcal{F}_B$ denotes a mixture (compounding) distribution, where $\mathcal{F}_A$ represents the original distribution and $\mathcal{F}_B$ the mixing distribution (i.e., the distribution of $\delta$) [15]; all vectors are row vectors, and $x^\top$ is the transposed of the row vector $x$; for any vector $x$, $x_k$ denotes its $k$th coordinate, and $\|x\|$ its Euclidean norm; $\mathbb{N}_0 = \{0, 1, 2, 3, \ldots\}$; $I\{A\}$ denotes the indicator function of the set $A$; $P_\theta$ denotes the probability law of the BHD with parameter $\theta$; $E_\theta$ denotes expectation with respect to the probability function $P_\theta$; $P_*$ and $E_*$ denotes the conditional probability law and expectation, given the data $(X_1, Y_1), \ldots, (X_n, Y_n)$, respectively; all limits in this work are taken as $n \to \infty$; $\xrightarrow{L}$ denotes convergence in distribution; $\xrightarrow{a.s.}$ denotes almost sure convergence; let $\{C_n\}$ be a sequence of random variables or random elements and let $\epsilon \in \mathbb{R}$; then, $C_n = O_p(n^{-\epsilon})$ means that $n^\epsilon C_n$ is bounded in probability, $C_n = o_p(n^{-\epsilon})$ means that $n^\epsilon C_n \xrightarrow{P} 0$ and $C_n = o(n^{-\epsilon})$ means that $n^\epsilon C_n \xrightarrow{a.s.} 0$ and $\mathcal{H} = L^2([0,1]^2, \varrho)$ denotes the separable Hilbert space of the measurable functions $\varphi, \varrho : [0,1]^2 \to \mathbb{R}$ such that $\|\varphi\|_\mathcal{H}^2 = \int_0^1 \int_0^1 \varphi^2(t)\, \varrho(t)\, dt < \infty$.

## 2. Preliminaries

Several definitions for the BHD have been given (see, for example, Kocherlakota and Kocherlakota in [16]). In this paper, we will work with the following one, which has received more attention in the statistical literature (see, for example, Papageorgiou et al. in [17]; Kemp et al. in [18]).

Let $\boldsymbol{X} = (X_1, X_2)$ have the bivariate Poisson distribution with the parameters $\delta\lambda_1, \delta\lambda_2$, and $\delta\lambda_3$ (for more details of this distribution; see, for example, Johnson et al. in [19]); then, $\boldsymbol{X} \underset{\delta}{\wedge} N(\mu, \sigma^2)$ has the BHD. Kocherlakota in [20] obtained its pgf, which is given by

$$v(t; \theta) = \exp\left(\mu\lambda + \frac{1}{2}\sigma^2\lambda^2\right), \tag{1}$$

where $t = (t_1, t_2)$, $\theta = (\mu, \sigma^2, \lambda_1, \lambda_2, \lambda_3)$, $\lambda = \lambda_1(t_1 - 1) + \lambda_2(t_2 - 1) + \lambda_3(t_1 t_2 - 1)$ and $\mu > \sigma^2(\lambda_i + \lambda_3)$, $i = 1, 2$.

From the pgf of the BHD, Kocherlakota and Kocherlakota [16] obtained the probability mass function of the BHD, which is given by

$$f(r, s) = \frac{\lambda_1^r \lambda_2^s}{r! s!} M(\gamma) \sum_{k=0}^{\min(r,s)} \binom{r}{k} \binom{s}{k} k! \, \xi^k P_{r+s-k}(\gamma),$$

where $M(x)$ is the moment-generating function of the normal distribution, $P_r(x)$ is a polynomial of degree $r$ in $x$, $\gamma = -(\lambda_1 + \lambda_2 + \lambda_3)$ and $\xi = \frac{\lambda_3}{\lambda_1 \lambda_2}$.

**Remark 1.** *If $\lambda_3 = 0$, then the probability function is reduced to*

$$f(r,s) = \frac{\lambda_1^r \lambda_2^s}{r! s!} M(-\lambda_1 - \lambda_2) P_{r+s}(-\lambda_1 - \lambda_2).$$

**Remark 2.** *If $X$ is a random vector that is bivariate Hermite distributed with parameter $\theta$, it will be denoted $X \sim BH(\theta)$, where $\theta \in \Theta$, and the parameter space is*

$$\Theta = \left\{ (\mu, \sigma^2, \lambda_1, \lambda_2, \lambda_3) \in \mathbb{R}^5 / \mu > \sigma^2(\lambda_i + \lambda_3), \lambda_i > \lambda_3 \geq 0, i = 1, 2 \right\}.$$

Let $X_1 = (X_{11}, X_{12}), X_2 = (X_{21}, X_{22}), \ldots, X_n = (X_{n1}, X_{n2})$ be independent and identically distributed (iid) random vectors defined on a probability space $(\Omega, \mathcal{A}, P)$ and taking values in $\mathbb{N}_0^2$. In what follows, let

$$v_n(t) = \frac{1}{n} \sum_{i=1}^{n} t_1^{X_{i1}} t_2^{X_{i2}}$$

denote the epgf of $X_1, X_2, \ldots, X_n$ for some appropriate $W \subseteq \mathbb{R}^2$.

The following section is dedicated to developing the statistic proposed in this study and, for this, it is essential to know the result that is presented below, the proof of which can be reviewed in [14]:

**Proposition 1.** *Let $X_1, \ldots, X_n$ be iid from a random vector $X = (X_1, X_2) \in \mathbb{N}_0^2$. Let $v(t) = E\left(t_1^{X_1} t_2^{X_2}\right)$ be the pgf of $X$, defined on $W \subseteq \mathbb{R}^2$. Let $0 \leq b_j \leq c_j < \infty$, $j = 1, 2$, such that $Q = [b_1, c_1] \times [b_2, c_2] \subseteq W$; then,*

$$\sup_{t \in Q} |v_n(t) - v(t)| \xrightarrow{a.s.} 0.$$

## 3. The Test Statistic and Its Asymptotic Null Distribution

Let $X_1 = (X_{11}, X_{12}), X_2 = (X_{21}, X_{22}), \ldots, X_n = (X_{n1}, X_{n2})$ be iid from a random vector $X = (X_1, X_2) \in \mathbb{N}_0^2$. Based on the sample $X_1, X_2, \ldots, X_n$, the objective is to test the hypothesis

$$H_0 : (X_1, X_2) \sim BH(\theta), \text{ for some } \theta \in \Theta,$$

against the alternative

$$H_1 : (X_1, X_2) \nsim BH(\theta), \forall \theta \in \Theta.$$

With this purpose, we will recourse to some of the properties of the pgf that allow us to propose the following statistical test.

According to Proposition 1, a consistent estimator of the pgf is the epgf. If $H_0$ is true and $\hat{\theta}_n$ is a consistent estimator of $\theta$, then $v(t; \hat{\theta}_n)$ consistently estimates the population pgf. Since the distribution of $X = (X_1, X_2)$ is uniquely determined by its pgf, $v(t)$, $t = (t_1, t_2) \in [0, 1]^2$, a reasonable test for testing $H_0$ should reject the null hypothesis for large values of $V_{n,w}(\hat{\theta}_n)$ defined by

$$V_{n,w}(\hat{\theta}_n) = \int_0^1 \int_0^1 V_n^2(t; \hat{\theta}_n) w(t) dt, \tag{2}$$

where

$$V_n(t; \theta) = \sqrt{n} \{ v_n(t) - v(t; \theta) \},$$

$\hat{\theta}_n = \hat{\theta}_n(X_1, X_2, \ldots, X_n)$ is a consistent estimator of $\theta$ and $w(t)$ is a measurable weight function, such that $w(t) \geq 0$, $\forall t \in [0, 1]^2$, and

$$\int_0^1 \int_0^1 w(t) dt < \infty. \tag{3}$$

The assumption (3) on $w$ ensures that the double integral in (2) is finite for each fixed $n$. Now, to determine what are large values of $V_{n,w}(\hat{\theta}_n)$, we must calculate its null distribution, or at least an approximation to it. Since the null distribution of $V_{n,w}(\hat{\theta}_n)$ is unknown, we first try to estimate it by means of its asymptotic null distribution. In order to derive it, we will assume that the estimator $\hat{\theta}_n$ satisfies the following regularity condition:

**Assumption 1.** *Under $H_0$, if $\theta = (\mu, \sigma^2, \lambda_1, \lambda_2, \lambda_3) \in \Theta$ denotes the true parameter value, then*

$$\sqrt{n}(\hat{\theta}_n - \theta) = \frac{1}{\sqrt{n}} \sum_{i=1}^{n} \boldsymbol{\ell}(\boldsymbol{X}_i; \theta) + \mathbf{o}_P(1),$$

*where $\boldsymbol{\ell} : \mathbb{N}_0^2 \times \Theta \longrightarrow \mathbb{R}^5$ is such that $E_\theta\{\boldsymbol{\ell}(\boldsymbol{X}_1; \theta)\} = \mathbf{0}$ and $J(\theta) = E_\theta\left\{\boldsymbol{\ell}(\boldsymbol{X}_1; \theta)^\top \boldsymbol{\ell}(\boldsymbol{X}_1; \theta)\right\} < \infty$.*

Assumption 1 is fulfilled by most commonly used estimators; see [16,21]. The next result gives the asymptotic null distribution of $V_{n,w}(\hat{\theta}_n)$.

**Theorem 1.** *Let $\boldsymbol{X}_1, \ldots, \boldsymbol{X}_n$ be iid from $\boldsymbol{X} = (X_1, X_2) \sim BH(\theta)$. Suppose that Assumption 1 holds. Then*

$$V_{n,w}(\hat{\theta}_n) = ||W_n||_{\mathcal{H}}^2 + o_P(1),$$

*where $W_n(t) = \frac{1}{\sqrt{n}} \sum_{i=1}^{n} V^0(\boldsymbol{X}_i, \theta; t)$, with*

$$V^0(\boldsymbol{X}_i, \theta; t) = t_1^{X_{i1}} t_2^{X_{i2}} - v(t; \theta)\left\{1 + \left(\lambda, \frac{1}{2}\lambda^2, \eta(t_1 - 1), \eta(t_2 - 1), \eta(t_1 t_2 - 1)\right)\boldsymbol{\ell}(\boldsymbol{X}_i; \theta)^\top\right\},$$

*$i = 1, \ldots, n$, $\eta = \mu + \sigma^2\lambda$. Moreover,*

$$V_{n,w}(\hat{\theta}_n) \xrightarrow{L} \sum_{j \geq 1} \lambda_j \chi_{1j}^2, \tag{4}$$

*where $\chi_{11}^2, \chi_{12}^2, \ldots$ are independent $\chi^2$ variates with one degree of freedom and the set $\{\lambda_j\}$ is the non-null eigenvalues of the operator $C(\theta)$ defined on the function space $\{\tau : \mathbb{N}_0^2 \to \mathbb{R}, \text{ such that } E_\theta\{\tau^2(\boldsymbol{X})\} < \infty, \forall \theta \in \Theta\}$, as follows:*

$$C(\theta)\tau(\boldsymbol{x}) = E_\theta\{h(\boldsymbol{x}, \boldsymbol{Y}; \theta)\tau(\boldsymbol{Y})\},$$

*where*

$$h(\boldsymbol{x}, \boldsymbol{y}; \theta) = \int_0^1 \int_0^1 V^0(\boldsymbol{x}; \theta; t) V^0(\boldsymbol{y}; \theta; t) w(t) dt. \tag{5}$$

**Proof.** By definition, $V_{n,w}(\hat{\theta}_n) = \|V_n(\hat{\theta}_n)\|_{\mathcal{H}}^2$. Note that

$$V_n(t; \hat{\theta}_n) = \frac{1}{\sqrt{n}} \sum_{i=1}^{n} V(\boldsymbol{X}_i; \hat{\theta}_n; t), \quad \text{with } V(\boldsymbol{X}_i; \theta; t) = t_1^{X_{i1}} t_2^{X_{i2}} - v(t; \theta). \tag{6}$$

By Taylor expansion of $V(\boldsymbol{X}_i; \hat{\theta}_n; t)$ around $\hat{\theta}_n = \theta$,

$$V_n(t; \hat{\theta}_n) = \frac{1}{\sqrt{n}} \sum_{i=1}^{n} V(\boldsymbol{X}_i; \theta; t) + \frac{1}{n} \sum_{i=1}^{n} Q^{(1)}(\boldsymbol{X}_i; \theta; t) \sqrt{n}(\hat{\theta}_n - \theta)^\top + q_n, \tag{7}$$

where $q_n = \frac{1}{2\sqrt{n}}(\hat{\theta}_n - \theta) \sum_{i=1}^{n} Q^{(2)}(\boldsymbol{X}_i; \tilde{\theta}; t)(\hat{\theta}_n - \theta)^\top$, $\tilde{\theta} = \alpha\hat{\theta}_n + (1 - \alpha)\theta$, for some $0 < \alpha < 1$, $Q^{(1)}(\boldsymbol{x}; \vartheta; t)$ is the vector of the first derivatives and $Q^{(2)}(\boldsymbol{x}; \vartheta; t)$ is the matrix of the second derivatives of $V(\boldsymbol{x}; \vartheta; t)$ with respect to $\vartheta$.

Thus, considering (3) results in

$$E_\theta\left\{\left\|Q_j^{(1)}(\boldsymbol{X}_1;\theta;t)\right\|_{\mathcal{H}}^2\right\} < \infty, \ \ j = 1, 2, \ldots, 5. \tag{8}$$

Using the Markov inequality and (8), we have

$$P_\theta\left[\left\|\frac{1}{n}\sum_{i=1}^n Q_j^{(1)}(\boldsymbol{X}_i;\theta;t) - E_\theta\left\{Q_j^{(1)}(\boldsymbol{X}_1;\theta;t)\right\}\right\|_{\mathcal{H}} > \varepsilon\right]$$

$$\leq \frac{1}{n\,\varepsilon^2}E_\theta\left[\left\|Q_j^{(1)}(\boldsymbol{X}_1;\theta;t)\right\|_{\mathcal{H}}^2\right] \to 0, \ \ j = 1, 2, \ldots, 5.$$

Then,

$$\frac{1}{n}\sum_{i=1}^n Q^{(1)}(\boldsymbol{X}_i;\theta;t) \xrightarrow{P} E_\theta\left\{Q^{(1)}(\boldsymbol{X}_1;\theta;t)\right\},$$

where $E_\theta\left\{Q^{(1)}(\boldsymbol{X}_1;\theta;t)\right\} = -v(t;\theta)\left(\lambda, \frac{1}{2}\lambda^2, \eta(t_1 - 1), \eta(t_2 - 1), \eta(t_1 t_2 - 1)\right)$.

As $\|q_n\|_{\mathcal{H}} = o_P(1)$, then, using Assumption 1, (7) can be written as

$$V_n(t;\hat{\theta}_n) = S_n(t;\theta) + s_n,$$

where $\|s_n\|_{\mathcal{H}} = o_P(1)$, and

$$S_n(t;\theta) = \frac{1}{\sqrt{n}}\sum_{i=1}^n\left[V(\boldsymbol{X}_i;\theta;t) + E_\theta\left\{Q^{(1)}(\boldsymbol{X}_1;\theta;t)\right\}\boldsymbol{\ell}(\boldsymbol{X}_i;\theta)^\top\right].$$

On the other hand, observe that

$$\|S_n(\theta)\|_{\mathcal{H}}^2 = \frac{1}{n}\sum_{i=1}^n\sum_{j=1}^n h(\boldsymbol{X}_i, \boldsymbol{X}_j;\theta),$$

where $h(\boldsymbol{x},\boldsymbol{y};\theta)$ is defined in (5) and satisfies $h(\boldsymbol{x},\boldsymbol{y};\theta) = h(\boldsymbol{y},\boldsymbol{x};\theta)$, $E_\theta\left\{h^2(\boldsymbol{X}_1,\boldsymbol{X}_2;\theta)\right\} < \infty$, $E_\theta\{|h(\boldsymbol{X}_1,\boldsymbol{X}_1;\theta)|\} < \infty$ and $E_\theta\{h(\boldsymbol{X}_1,\boldsymbol{X}_2;\theta)\} = 0$. Thus, from Theorem 6.4.1.B in Serfling [22],

$$\|S_n(\theta)\|_{\mathcal{H}}^2 \xrightarrow{L} \sum_{j\geq 1}\lambda_j \chi_{1j}^2,$$

where $\chi_{11}^2, \chi_{12}^2, \ldots$ and the set $\{\lambda_j\}$ are as defined in the statement of the Theorem. In particular, $\|S_n(\theta)\|_{\mathcal{H}}^2 = O_P(1)$, which implies (4). $\quad\square$

The asymptotic null distribution of $V_{n,w}(\hat{\theta}_n)$ depends on the unknown true value of the parameter $\theta$; therefore, in practice, they do not provide a useful solution to the problem of estimating the null distribution of the respective statistical tests. This could be solved by replacing $\theta$ with $\hat{\theta}$.

However, a greater difficulty is to determine the sets $\{\lambda_j\}_{j\geq 1}$; for most of the cases, calculating the eigenvalues of an operator is not a simple task and, in our case, we must also obtain the expression $h(\boldsymbol{x},\boldsymbol{y};\theta)$, which is not easy to find, since it depends on the function $\boldsymbol{\ell}$, which usually does not have a simple expression.

Thus, in the next section, we consider another way to approximate the null distribution of the statistical test, the parametric bootstrap method.

## 4. The Bootstrap Estimator

An alternative way to estimate the null distribution is through the parametric bootstrap method.

Let $X_1, \ldots, X_n$ be iid taking values in $\mathbb{N}_0^2$. Assume that $\hat{\theta}_n = \hat{\theta}_n(X_1, \ldots, X_n) \in \Theta$. Let $X_1^*, \ldots, X_n^*$ be iid from a population with distribution $BH(\hat{\theta}_n)$, given $X_1, \ldots, X_n$, and let $V_{n,w}^*(\hat{\theta}_n^*)$ be the bootstrap version of $V_{n,w}(\hat{\theta}_n)$ obtained by replacing $X_1, \ldots, X_n$ and $\hat{\theta}_n = \hat{\theta}_n(X_1, \ldots, X_n)$ by $X_1^*, \ldots, X_n^*$ and $\hat{\theta}_n^* = \hat{\theta}_n(X_1^*, \ldots, X_n^*)$, respectively, in the expression of $V_{n,w}(\hat{\theta}_n)$. Let $P_*$ denote the bootstrap conditional probability law, given $X_1, \ldots, X_n$. In order to show that the bootstrap consistently estimate the null distribution of $V_{n,w}(\hat{\theta}_n)$, we will assume the following assumption, which is a bit stronger than Assumption 1.

**Assumption 2.** *Assumption 1 holds and the functions $\ell$ and $J$ satisfy*
(1)    $\sup_{\vartheta \in \Theta_0} E_\vartheta\big[\|\boldsymbol{\ell}(X; \vartheta)\|^2 I\{\|\boldsymbol{\ell}(X; \vartheta)\| > \gamma\}\big] \longrightarrow 0$, *as $\gamma \to \infty$, where $\Theta_0 \subseteq \Theta$ is an open neighborhood of $\theta$.*
(2)    $\boldsymbol{\ell}(X; \vartheta)$ *is continuous as a function of $\vartheta$ at $\vartheta = \theta$, and $J(\vartheta)$ is finite $\forall \vartheta \in \Theta_0$.*

As stated after Assumption 1, Assumption 2 is not restrictive since it is fulfilled by commonly used estimators.

The next theorem shows that the bootstrap distribution of $V_{n,w}(\hat{\theta}_n)$ consistently estimates its null distribution.

**Theorem 2.** *Let $X_1, \ldots, X_n$ be iid from a random vector $X = (X_1, X_2) \in \mathbb{N}_0^2$. Suppose that Assumption 2 holds and that $\hat{\theta}_n = \theta + o(1)$, for some $\theta \in \Theta$. Then,*

$$\sup_{x \in \mathbb{R}} \big| P_*\big\{V_{n,w}^*(\hat{\theta}_n^*) \le x\big\} - P_\theta\big\{V_{n,w}(\hat{\theta}_n) \le x\big\}\big| \xrightarrow{a.s.} 0.$$

**Proof.** By definition, $V_{n,w}^*(\hat{\theta}_n^*) = \|V_n^*(\hat{\theta}_n^*)\|_{\mathcal{H}}^2$, with

$$V_n^*(t; \hat{\theta}_n^*) = \frac{1}{\sqrt{n}} \sum_{i=1}^n V(X_i^*; \hat{\theta}_n^*; t)$$

and $V(X; \theta; t)$ defined in (6).

Following similar steps to those given in the proof of Theorem 1, it can be seen that $V_{n,w}^*(\hat{\theta}_n^*) = \|W_n^*\|_{\mathcal{H}}^2 + o_{P_*}(1)$, where $W_n^*(t)$ is defined as $W_n(t)$ with $X_i$ and $\theta$ replaced by $X_i^*$ and $\hat{\theta}_n$, respectively.

To derive the result, first we will check that assumptions (i)–(iii) in Theorem 1.1 of Kundu et al. [23] hold.

Observe that

$$Y_n^*(t) = \sum_{i=1}^n Y_{ni}^*(t)$$

where

$$Y_{ni}^*(t) = \frac{1}{\sqrt{n}} V^0(X_i^*; \hat{\theta}_n; t), \quad i = 1, \ldots, n,$$

Clearly, $E_*\{Y_{ni}^*\} = 0$ and $E_*\{\|Y_{ni}^*\|_{\mathcal{H}}^2\} < \infty$. Let $K_n$ be the covariance kernel of $Y_n^*$, which by SLLN satisfies

$$
\begin{aligned}
K_n(u, v) &= E_*\{Y_n^*(u) Y_n^*(v)\} \\
&= E_*\Big\{V^0(X_1^*; \hat{\theta}_n; u) V^0(X_1^*; \hat{\theta}_n; v)\Big\} \\
&\xrightarrow{a.s.} E_\theta\Big\{V^0(X_1; \theta; u) V^0(X_1; \theta; v)\Big\} = K(u, v).
\end{aligned}
$$

Moreover, let $\mathcal{Z}$ be a zero-mean Gaussian process on $\mathcal{H}$ whose operator of covariance $C$ is characterized by

$$
\begin{aligned}
\langle Cf, h \rangle_{\mathcal{H}} &= cov(\langle \mathcal{Z}, f \rangle_{\mathcal{H}}, \langle \mathcal{Z}, h \rangle_{\mathcal{H}}) \\
&= \int_{[0,1]^4} K(u,v) f(u) h(v) w(u) w(v) du dv.
\end{aligned}
$$

From the central limit theorem in Hilbert spaces (see, for example, van der Vaart and Wellner [24]), it follows that $Y_n = \frac{1}{\sqrt{n}} \sum_{i=1}^{n} V^0(\boldsymbol{X}_i; \theta; t) \xrightarrow{L} \mathcal{Z}$ on $\mathcal{H}$, when the data are iid from the random vector $\boldsymbol{X} \sim HB(\theta)$.

Let $C_n$ denote the covariance operator of $Y_n^*$ and let $\{e_k : k \geq 0\}$ be an orthonormal basis of $\mathcal{H}$. Let $f, h \in \mathcal{H}$, by a dominated convergence theorem,

$$
\begin{aligned}
\lim_{n \to \infty} \langle C_n e_k, e_l \rangle_{\mathcal{H}} &= \lim_{n \to \infty} \int_{[0,1]^4} K_n(u,v) e_k(u) e_l(v) w(u) w(v) du dv \\
&= \langle C e_k, e_l \rangle_{\mathcal{H}}.
\end{aligned}
$$

Setting $a_{kl} = \langle C e_k, e_l \rangle_{\mathcal{H}}$ in the aforementioned Theorem 1.1, this proves that condition (i) holds. To verify condition (ii), by using a monotone convergence theorem, Parseval's relation and dominated convergence theorem, we obtained

$$
\begin{aligned}
\lim_{n \to \infty} \sum_{k=0}^{\infty} \langle C_n e_k, e_k \rangle_{\mathcal{H}} &= \lim_{n \to \infty} \sum_{k=0}^{\infty} \int_{[0,1]^4} K_n(u,v) e_k(u) e_k(v) w(u) w(v) du dv \\
&= \sum_{k=0}^{\infty} \int_{[0,1]^4} K(u,v) e_k(u) e_k(v) w(u) w(v) du dv = \sum_{k=0}^{\infty} \langle C e_k, e_k \rangle_{\mathcal{H}} \\
&= \sum_{k=0}^{\infty} a_{kk} = \sum_{k=0}^{\infty} E_\theta \left\{ \langle \mathcal{Z}, e_k \rangle_{\mathcal{H}_1}^2 \right\} = E_\theta \left\{ \| \mathcal{Z} \|_{\mathcal{H}}^2 \right\} < \infty.
\end{aligned}
$$

To prove condition (iii), we first notice that

$$
|\langle Y_{ni}^*, e_k \rangle_{\mathcal{H}}| \leq \frac{M}{\sqrt{n}}, \ i = 1, \ldots, n, \ \forall n, \text{ where } 0 < M < \infty.
$$

From the above inequality, for each fixed $\varepsilon > 0$,

$$
E_* \left[ \langle Y_{ni}^*, e_k \rangle_{\mathcal{H}}^2 \ I\{ |\langle Y_{ni}^*, e_k \rangle_{\mathcal{H}}| > \varepsilon \} \right] = 0.
$$

for sufficiently large $n$. This proves condition (iii). Therefore, $Y_n^* \xrightarrow{L} \mathcal{Z}$ in $\mathcal{H}$, a.s. Now, the result follows from the continuous mapping theorem. $\square$

From Theorem 2, the test function

$$
\Psi_V^* = \begin{cases} 1, & \text{if } V_{n,w}^*(\hat{\theta}_n^*) \geq v_{n,w,\alpha}^*, \\ 0, & \text{otherwise}, \end{cases}
$$

or, equivalently, the test that rejects $H_0$ when $p^* = P_*\{V_{n,w}^*(\hat{\theta}_n^*) \geq V_{obs}\} \leq \alpha$, is asymptotically correct in the sense that, when $H_0$ is true, $\lim P_\theta(\Psi_V^* = 1) = \alpha$, where $v_{n,w,\alpha}^* = \inf\{x : P_*(V_{n,w}^*(\hat{\theta}_n^*) \geq x) \leq \alpha\}$ is the $\alpha$ upper percentile of the bootstrap distribution of $V_{n,w}(\hat{\theta}_n)$ and $V_{obs}$ is the observed value of the test statistic.

## 5. Numerical Results and Discussion

According to Novoa-Muñoz and Jiménez-Gamero in [14], the properties of the statistic $V_{n,w}(\hat{\theta}_n)$ are asymptotic, that is, such properties describe the behavior of the test proposed for large samples. To study the goodness of the bootstrap approach for samples of finite

size, a simulation experiment was carried out. In this section, we describe this experiment and provide a summary of the results that have been obtained.

It is necessary to emphasize, as mentioned in the Introduction that, to the best of our knowledge, we have not found another goodness-of-fit test for the bivariate Hermite distribution with which we can make a comparison. Therefore, the simulation study is limited only to the test presented in this investigation.

On the other hand, all the computational calculations made in this paper were carried out through codes written in the R language [25].

To calculate $V_{n,w}(\hat{\theta}_n)$, it is necessary to give an explicit form to the weight function $w$. Here, the following is taken into account:

$$w(t; a_1, a_2) = t_1^{a_1} t_2^{a_2}. \tag{9}$$

Observe that the only restrictions that have been imposed on the weight function are that $w$ be positive almost everywhere in $[0,1]^2$ and the established in (3). The function $w(t; a_1, a_2)$ given in (9) meets these conditions whenever $a_i > -1, i = 1, 2$. Hence,

$$V_{n,w}(\hat{\theta}_n) = n \int_0^1 \int_0^1 \left[ \sum_{i=1}^n t_1^{X_{i1}} t_2^{X_{i2}} - exp\left( \hat{\mu}\hat{\lambda} + \frac{1}{2}\hat{\sigma}^2\hat{\lambda}^2 \right) \right]^2 t_1^{a_1} t_2^{a_2} \, dt_1 dt_2.$$

It was not possible to find an explicit form of the statistic $V_{n,w}(\hat{\theta}_n)$, for which its calculation used the curvature package of R [25] to calculate it.

### 5.1. Simulated Data

In order to approximate the null distribution of the statistic $V_{n,w}(\hat{\theta}_n)$ for finite-size samples of sizes 30, 50, and 70 from a $BH(\theta)$, for $\theta = (\mu, \sigma^2, \lambda_1, \lambda_2, \lambda_3)$, the pgf (1), with $\lambda_3 = 0$, was utilized. The combinations of parameters were chosen in such a way that $\mu > \sigma^2(\lambda_i + \lambda_3), i = 1, 2$.

The selected values of the other parameters were $\mu \in \{1.0, 1.5, 2.0\}$, $\sigma^2 \in \{0.8, 1.0\}$, $\lambda_1 \in \{0.10, 0.25, 0.50, 0.75, 1.00\}$ and $\lambda_2 \in \{0.20, 0.25, 0.50, 0.75\}$.

The selected values of $\lambda_1$ and $\lambda_2$ were not greater than 1 since the Hermite distribution is characterized as being zero-inflated.

To estimate the parameter $\theta$, we use the maximum likelihood method given in Kocherlakota and Kocherlakota [16]. Then, we approximated the bootstrap $p$-values of the proposed test with the weight function given in (9) for $(a_1, a_2) \in \{(0, 0), (1, 0), (0, 1), (1, 1), (5, 1), (1, 5), (5, 5)\}$, and we generate $B = 500$ bootstrap samples.

The above procedure was repeated 1000 times, and the fraction of the estimated $p$-values that was found to be less than or equal to 0.05 and 0.10, which are the estimates type I error probabilities for $\alpha = 0.05$ and 0.1.

The results obtained are presented in Tables 1–7 for the different pairs $(a_1, a_2)$. In each table, the established order was growing in $\mu$ and $\sigma^2$, and for each new $\mu$ increasing values in $\lambda_1$, and in each new $\lambda_1$, increasing values for $\lambda_2$. From these results, we can conclude that the parametric bootstrap method provides good approximations to the null distribution of the $V_{n,w}(\hat{\theta}_n)$ in most of the cases considered.

It is seen that the values of $a_1$ and $a_2$ of the weight function affect bootstrap estimates of $p$-values.

From the tables, it is clear that the bootstrap $p$-values are increasingly approaching the nominal value as $n$ increases. These approximations are better when $a_1 = a_2$. In particular, when $a_1 = a_2$ is small (less than 5), then the bootstrap $p$-values are approached from the left (below) to the nominal value; otherwise, it happens when $a_1 = a_2$ are fairly large values (greater or equal to 5). Table 4 is the one that shows the best results, being the weight function with $a_1 = a_2 = 1$ that presents the best $p$-values estimates.

**Table 1.** Simulation results for the probability of type I error for $a_1 = 0$ and $a_2 = 0$.

| $\theta$ | $n = 30$ | | $n = 50$ | | $n = 70$ | |
|---|---|---|---|---|---|---|
| | $\alpha = 0.05$ | $\alpha = 0.1$ | $\alpha = 0.05$ | $\alpha = 0.1$ | $\alpha = 0.05$ | $\alpha = 0.1$ |
| (1.0, 0.8, 0.10, 0.20, 0.00) | 0.012 | 0.053 | 0.029 | 0.069 | 0.037 | 0.081 |
| (1.0, 0.8, 0.25, 0.25, 0.00) | 0.027 | 0.067 | 0.037 | 0.064 | 0.043 | 0.094 |
| (1.0, 0.8, 0.50, 0.20, 0.00) | 0.016 | 0.062 | 0.046 | 0.073 | 0.047 | 0.087 |
| (1.0, 0.8, 0.50, 0.50, 0.00) | 0.025 | 0.063 | 0.042 | 0.076 | 0.044 | 0.091 |
| (1.5, 1.0, 0.50, 0.50, 0.00) | 0.010 | 0.064 | 0.035 | 0.078 | 0.042 | 0.089 |
| (1.5, 1.0, 0.50, 0.75, 0.00) | 0.010 | 0.065 | 0.036 | 0.084 | 0.041 | 0.084 |
| (1.5, 1.0, 0.75, 0.25, 0.00) | 0.017 | 0.071 | 0.038 | 0.087 | 0.043 | 0.088 |
| (1.5, 1.0, 1.00, 0.25, 0.00) | 0.027 | 0.076 | 0.039 | 0.090 | 0.042 | 0.092 |
| (2.0, 1.0, 0.25, 0.75, 0.00) | 0.017 | 0.067 | 0.038 | 0.082 | 0.047 | 0.089 |
| (2.0, 1.0, 0.50, 0.25, 0.00) | 0.011 | 0.067 | 0.037 | 0.088 | 0.045 | 0.091 |
| (2.0, 1.0, 0.75, 0.25, 0.00) | 0.029 | 0.070 | 0.035 | 0.087 | 0.043 | 0.089 |

**Table 2.** Simulation results for the probability of type I error for $a_1 = 1$ and $a_2 = 0$.

| $\theta$ | $n = 30$ | | $n = 50$ | | $n = 70$ | |
|---|---|---|---|---|---|---|
| | $\alpha = 0.05$ | $\alpha = 0.1$ | $\alpha = 0.05$ | $\alpha = 0.1$ | $\alpha = 0.05$ | $\alpha = 0.1$ |
| (1.0, 0.8, 0.10, 0.20, 0.00) | 0.010 | 0.039 | 0.025 | 0.073 | 0.043 | 0.088 |
| (1.0, 0.8, 0.25, 0.25, 0.00) | 0.025 | 0.073 | 0.037 | 0.088 | 0.041 | 0.104 |
| (1.0, 0.8, 0.50, 0.20, 0.00) | 0.027 | 0.072 | 0.041 | 0.083 | 0.045 | 0.086 |
| (1.0, 0.8, 0.50, 0.50, 0.00) | 0.035 | 0.053 | 0.042 | 0.072 | 0.045 | 0.101 |
| (1.5, 1.0, 0.50, 0.50, 0.00) | 0.011 | 0.064 | 0.031 | 0.080 | 0.038 | 0.085 |
| (1.5, 1.0, 0.50, 0.75, 0.00) | 0.019 | 0.065 | 0.034 | 0.078 | 0.039 | 0.080 |
| (1.5, 1.0, 0.75, 0.25, 0.00) | 0.025 | 0.081 | 0.038 | 0.085 | 0.042 | 0.084 |
| (1.5, 1.0, 1.00, 0.25, 0.00) | 0.037 | 0.074 | 0.035 | 0.085 | 0.040 | 0.086 |
| (2.0, 1.0, 0.25, 0.75, 0.00) | 0.027 | 0.071 | 0.034 | 0.082 | 0.047 | 0.089 |
| (2.0, 1.0, 0.50, 0.25, 0.00) | 0.011 | 0.077 | 0.031 | 0.084 | 0.044 | 0.086 |
| (2.0, 1.0, 0.75, 0.25, 0.00) | 0.019 | 0.080 | 0.035 | 0.085 | 0.044 | 0.087 |

**Table 3.** Simulation results for the probability of type I error for $a_1 = 0$ and $a_2 = 1$.

| $\theta$ | $n = 30$ | | $n = 50$ | | $n = 70$ | |
|---|---|---|---|---|---|---|
| | $\alpha = 0.05$ | $\alpha = 0.1$ | $\alpha = 0.05$ | $\alpha = 0.1$ | $\alpha = 0.05$ | $\alpha = 0.1$ |
| (1.0, 0.8, 0.10, 0.20, 0.00) | 0.014 | 0.044 | 0.029 | 0.067 | 0.043 | 0.088 |
| (1.0, 0.8, 0.25, 0.25, 0.00) | 0.028 | 0.068 | 0.039 | 0.079 | 0.042 | 0.084 |
| (1.0, 0.8, 0.50, 0.20, 0.00) | 0.019 | 0.063 | 0.042 | 0.083 | 0.057 | 0.092 |
| (1.0, 0.8, 0.50, 0.50, 0.00) | 0.029 | 0.063 | 0.045 | 0.075 | 0.054 | 0.089 |
| (1.5, 1.0, 0.50, 0.50, 0.00) | 0.011 | 0.066 | 0.039 | 0.079 | 0.042 | 0.089 |
| (1.5, 1.0, 0.50, 0.75, 0.00) | 0.013 | 0.070 | 0.043 | 0.082 | 0.043 | 0.087 |
| (1.5, 1.0, 0.75, 0.25, 0.00) | 0.017 | 0.081 | 0.042 | 0.089 | 0.043 | 0.092 |
| (1.5, 1.0, 1.00, 0.25, 0.00) | 0.037 | 0.086 | 0.045 | 0.091 | 0.045 | 0.093 |
| (2.0, 1.0, 0.25, 0.75, 0.00) | 0.047 | 0.077 | 0.048 | 0.084 | 0.047 | 0.089 |
| (2.0, 1.0, 0.50, 0.25, 0.00) | 0.014 | 0.077 | 0.037 | 0.089 | 0.043 | 0.093 |
| (2.0, 1.0, 0.75, 0.25, 0.00) | 0.027 | 0.080 | 0.041 | 0.097 | 0.044 | 0.096 |

**Table 4.** Simulation results for the probability of type I error for $a_1 = 1$ and $a_2 = 1$.

| $\theta$ | $n = 30$ | | $n = 50$ | | $n = 70$ | |
|---|---|---|---|---|---|---|
| | $\alpha = 0.05$ | $\alpha = 0.1$ | $\alpha = 0.05$ | $\alpha = 0.1$ | $\alpha = 0.05$ | $\alpha = 0.1$ |
| (1.0, 0.8, 0.10, 0.20, 0.00) | 0.016 | 0.073 | 0.024 | 0.086 | 0.048 | 0.092 |
| (1.0, 0.8, 0.25, 0.25, 0.00) | 0.032 | 0.058 | 0.037 | 0.088 | 0.049 | 0.091 |
| (1.0, 0.8, 0.50, 0.20, 0.00) | 0.024 | 0.064 | 0.043 | 0.085 | 0.048 | 0.089 |
| (1.0, 0.8, 0.50, 0.50, 0.00) | 0.033 | 0.072 | 0.043 | 0.086 | 0.049 | 0.093 |
| (1.5, 1.0, 0.50, 0.50, 0.00) | 0.030 | 0.072 | 0.038 | 0.088 | 0.046 | 0.090 |
| (1.5, 1.0, 0.50, 0.75, 0.00) | 0.033 | 0.071 | 0.042 | 0.084 | 0.047 | 0.098 |
| (1.5, 1.0, 0.75, 0.25, 0.00) | 0.036 | 0.097 | 0.039 | 0.097 | 0.049 | 0.099 |
| (1.5, 1.0, 1.00, 0.25, 0.00) | 0.039 | 0.088 | 0.046 | 0.090 | 0.049 | 0.093 |
| (2.0, 1.0, 0.25, 0.75, 0.00) | 0.031 | 0.087 | 0.044 | 0.092 | 0.048 | 0.099 |
| (2.0, 1.0, 0.50, 0.25, 0.00) | 0.035 | 0.068 | 0.039 | 0.081 | 0.047 | 0.093 |
| (2.0, 1.0, 0.75, 0.25, 0.00) | 0.037 | 0.080 | 0.045 | 0.088 | 0.049 | 0.096 |

**Table 5.** Simulation results for the probability of type I error for $a_1 = 1$ and $a_2 = 5$.

| $\theta$ | $n = 30$ | | $n = 50$ | | $n = 70$ | |
|---|---|---|---|---|---|---|
| | $\alpha = 0.05$ | $\alpha = 0.1$ | $\alpha = 0.05$ | $\alpha = 0.1$ | $\alpha = 0.05$ | $\alpha = 0.1$ |
| (1.0, 0.8, 0.10, 0.20, 0.00) | 0.014 | 0.037 | 0.032 | 0.075 | 0.051 | 0.093 |
| (1.0, 0.8, 0.25, 0.25, 0.00) | 0.023 | 0.074 | 0.053 | 0.090 | 0.060 | 0.113 |
| (1.0, 0.8, 0.50, 0.20, 0.00) | 0.036 | 0.101 | 0.062 | 0.110 | 0.064 | 0.117 |
| (1.0, 0.8, 0.50, 0.50, 0.00) | 0.023 | 0.080 | 0.042 | 0.107 | 0.063 | 0.109 |
| (1.5, 1.0, 0.50, 0.50, 0.00) | 0.022 | 0.081 | 0.037 | 0.111 | 0.046 | 0.108 |
| (1.5, 1.0, 0.50, 0.75, 0.00) | 0.039 | 0.095 | 0.048 | 0.108 | 0.056 | 0.108 |
| (1.5, 1.0, 0.75, 0.25, 0.00) | 0.034 | 0.108 | 0.048 | 0.107 | 0.054 | 0.108 |
| (1.5, 1.0, 1.00, 0.25, 0.00) | 0.037 | 0.107 | 0.059 | 0.109 | 0.054 | 0.107 |
| (2.0, 1.0, 0.25, 0.75, 0.00) | 0.048 | 0.106 | 0.056 | 0.108 | 0.054 | 0.106 |
| (2.0, 1.0, 0.50, 0.25, 0.00) | 0.025 | 0.107 | 0.047 | 0.108 | 0.045 | 0.108 |
| (2.0, 1.0, 0.75, 0.25, 0.00) | 0.043 | 0.107 | 0.045 | 0.107 | 0.043 | 0.106 |

**Table 6.** Simulation results for the probability of type I error for $a_1 = 5$ and $a_2 = 1$.

| $\theta$ | $n = 30$ | | $n = 50$ | | $n = 70$ | |
|---|---|---|---|---|---|---|
| | $\alpha = 0.05$ | $\alpha = 0.1$ | $\alpha = 0.05$ | $\alpha = 0.1$ | $\alpha = 0.05$ | $\alpha = 0.1$ |
| (1.0, 0.8, 0.10, 0.20, 0.00) | 0.015 | 0.040 | 0.032 | 0.062 | 0.042 | 0.081 |
| (1.0, 0.8, 0.25, 0.25, 0.00) | 0.034 | 0.076 | 0.045 | 0.101 | 0.048 | 0.104 |
| (1.0, 0.8, 0.50, 0.20, 0.00) | 0.028 | 0.084 | 0.048 | 0.073 | 0.053 | 0.089 |
| (1.0, 0.8, 0.50, 0.50, 0.00) | 0.028 | 0.069 | 0.045 | 0.079 | 0.054 | 0.098 |
| (1.5, 1.0, 0.50, 0.50, 0.00) | 0.019 | 0.071 | 0.035 | 0.078 | 0.042 | 0.099 |
| (1.5, 1.0, 0.50, 0.75, 0.00) | 0.044 | 0.104 | 0.048 | 0.098 | 0.056 | 0.104 |
| (1.5, 1.0, 0.75, 0.25, 0.00) | 0.027 | 0.107 | 0.038 | 0.105 | 0.046 | 0.103 |
| (1.5, 1.0, 1.00, 0.25, 0.00) | 0.037 | 0.117 | 0.043 | 0.112 | 0.060 | 0.107 |
| (2.0, 1.0, 0.25, 0.75, 0.00) | 0.037 | 0.112 | 0.039 | 0.108 | 0.054 | 0.108 |
| (2.0, 1.0, 0.50, 0.25, 0.00) | 0.026 | 0.077 | 0.034 | 0.109 | 0.055 | 0.109 |
| (2.0, 1.0, 0.75, 0.25, 0.00) | 0.034 | 0.116 | 0.045 | 0.107 | 0.056 | 0.105 |

**Table 7.** Simulation results for the probability of type I error for $a_1 = 5$ and $a_2 = 5$.

| $\theta$ | $n = 30$ | | $n = 50$ | | $n = 70$ | |
|---|---|---|---|---|---|---|
| | $\alpha = 0.05$ | $\alpha = 0.1$ | $\alpha = 0.05$ | $\alpha = 0.1$ | $\alpha = 0.05$ | $\alpha = 0.1$ |
| (1.0, 0.8, 0.10, 0.20, 0.00) | 0.017 | 0.035 | 0.032 | 0.065 | 0.050 | 0.089 |
| (1.0, 0.8, 0.25, 0.25, 0.00) | 0.027 | 0.077 | 0.034 | 0.081 | 0.043 | 0.084 |
| (1.0, 0.8, 0.50, 0.20, 0.00) | 0.030 | 0.086 | 0.042 | 0.087 | 0.048 | 0.104 |
| (1.0, 0.8, 0.50, 0.50, 0.00) | 0.013 | 0.069 | 0.030 | 0.076 | 0.045 | 0.105 |
| (1.5, 1.0, 0.50, 0.50, 0.00) | 0.016 | 0.063 | 0.035 | 0.078 | 0.046 | 0.087 |
| (1.5, 1.0, 0.50, 0.75, 0.00) | 0.019 | 0.085 | 0.061 | 0.089 | 0.054 | 0.094 |
| (1.5, 1.0, 0.75, 0.25, 0.00) | 0.031 | 0.071 | 0.053 | 0.102 | 0.047 | 0.098 |
| (1.5, 1.0, 1.00, 0.25, 0.00) | 0.037 | 0.086 | 0.049 | 0.104 | 0.052 | 0.102 |
| (2.0, 1.0, 0.25, 0.75, 0.00) | 0.015 | 0.087 | 0.057 | 0.098 | 0.055 | 0.101 |
| (2.0, 1.0, 0.75, 0.25, 0.00) | 0.040 | 0.097 | 0.054 | 0.102 | 0.053 | 0.102 |

Unfortunately, we could not find a closed form for our statistic $V_{n,w}(\hat{\theta}_n)$; in order to calculate it, we used the curvature package of the software R [25]. This had a serious impact on the computation time since the simulations were increased in their execution time by at least 30%.

### 5.2. The Power of a Hypothesis Test

To study the power, we repeated the previous experiment for samples of size $n = 50$ and, for the weight function, we used the values of $a_1$ and $a_2$ that yielded the best results in the study of type I error. The alternative distributions we use are detailed below:

- bivariate binomial distribution $BB(m; p_1, p_2, p_3)$, where $p_1 + p_2 - p_3 \leq 1$, $p_1 \geq p_3$, $p_2 \geq p_3$ and $p_3 > 0$,
- bivariate Poisson distribution $BP(\lambda_1, \lambda_2, \lambda_3)$, where $\lambda_1 > \lambda_3$, $\lambda_2 > \lambda_3 > 0$,
- bivariate logarithmic series distribution $BLS(\lambda_1, \lambda_2, \lambda_3)$, where $0 < \lambda_1 + \lambda_2 + \lambda_3 < 1$,

- bivariate negative binomial distribution $BNB(\nu; \gamma_0, \gamma_1, \gamma_2)$, where $\nu \in \mathbb{N}$, $\gamma_0 > \gamma_2$, $\gamma_1 > \gamma_2$ and $\gamma_2 > 0$,
- bivariate Neyman type A distribution $BNTA(\lambda; \lambda_1, \lambda_2, \lambda_3)$, where $0 < \lambda_1 + \lambda_2 + \lambda_3 \leq 1$,
- bivariate Poisson distribution mixtures of the form $pBP(\theta) + (1 - p)BP(\lambda)$, where $0 < p < 1$, denoted by $BPP(p; \theta, \lambda)$.

Table 8 displays the alternatives considered and the estimated power for nominal significance level $\alpha = 0.05$. Analyzing this table, we can conclude that all the considered tests, denoted by $V_{(a_1,a_2)}$, are able to detect the alternatives studied and with a good power, giving better results in cases where $a_1 = a_2$. The best result was achieved for $a_1 = a_2 = 1$, as expected, as occurred in the study of type I error.

**Table 8.** Simulation results for the power. The values are in the form of percentages, rounded to the nearest integer.

| Alternative | $V_{(0,0)}$ | $V_{(1,0)}$ | $V_{(1,1)}$ | $V_{(1,5)}$ | $V_{(5,5)}$ |
|---|---|---|---|---|---|
| $BB(1; 0.41, 0.02, 0.01)$ | 87 | 81 | 89 | 81 | 85 |
| $BB(1; 0.41, 0.03, 0.02)$ | 85 | 82 | 88 | 80 | 86 |
| $BB(2; 0.61, 0.01, 0.01)$ | 93 | 84 | 98 | 83 | 92 |
| $BB(1; 0.61, 0.03, 0.02)$ | 95 | 89 | 100 | 87 | 95 |
| $BB(2; 0.71, 0.01, 0.01)$ | 94 | 86 | 100 | 85 | 93 |
| $BP(1.00, 1.00, 0.25)$ | 85 | 76 | 89 | 77 | 82 |
| $BP(1.00, 1.00, 0.50)$ | 84 | 77 | 91 | 72 | 85 |
| $BP(1.00, 1.00, 0.75)$ | 87 | 75 | 92 | 73 | 83 |
| $BP(1.50, 1.00, 0.31)$ | 87 | 77 | 93 | 75 | 87 |
| $BP(1.50, 1.00, 0.92)$ | 86 | 76 | 92 | 77 | 87 |
| $BLS(0.25, 0.15, 0.10)$ | 94 | 85 | 98 | 86 | 95 |
| $BLS(5d/7, d/7, d/7)$ * | 91 | 85 | 100 | 84 | 90 |
| $BLS(3d/4, d/8, d/8)$ * | 90 | 86 | 100 | 84 | 90 |
| $BLS(7d/9, d/9, d/9)$ * | 94 | 86 | 100 | 83 | 93 |
| $BLS(0.51, 0.01, 0.02)$ | 90 | 83 | 98 | 83 | 91 |
| $BNB(1; 0.92, 0.97, 0.01)$ | 93 | 87 | 96 | 85 | 92 |
| $BNB(1; 0.97, 0.97, 0.01)$ | 92 | 86 | 95 | 85 | 92 |
| $BNB(1; 0.97, 0.97, 0.02)$ | 94 | 88 | 100 | 89 | 93 |
| $BNB(1; 0.98, 0.98, 0.01)$ | 92 | 84 | 97 | 85 | 92 |
| $BNB(1; 0.99, 0.99, 0.01)$ | 91 | 84 | 96 | 83 | 91 |
| $BNTA(0.21; 0.01, 0.01, 0.98)$ | 93 | 86 | 98 | 85 | 92 |
| $BNTA(0.24; 0.01, 0.01, 0.98)$ | 95 | 87 | 100 | 85 | 95 |
| $BNTA(0.26; 0.01, 0.01, 0.97)$ | 93 | 85 | 97 | 86 | 93 |
| $BNTA(0.26; 0.01, 0.01, 0.98)$ | 94 | 85 | 98 | 86 | 94 |
| $BNTA(0.28; 0.01, 0.01, 0.97)$ | 93 | 86 | 96 | 86 | 94 |
| $BPP(0.31; (0.2, 0.2, 0.1), (1.0, 1.0, 0.9))$ | 76 | 70 | 82 | 72 | 77 |
| $BPP(0.31; (0.2, 0.2, 0.1), (1.0, 1.2, 0.9))$ | 77 | 71 | 84 | 71 | 76 |
| $BPP(0.32; (0.2, 0.2, 0.1), (1.0, 1.0, 0.9))$ | 78 | 71 | 84 | 71 | 76 |
| $BPP(0.33; (0.2, 0.2, 0.1), (1.0, 1.0, 0.9))$ | 78 | 70 | 85 | 70 | 77 |
| $BPP(0.33; (0.2, 0.2, 0.1), (1.0, 1.1, 0.9))$ | 76 | 71 | 83 | 70 | 78 |

* $d = 1 - \exp(-1) \approx 0.63212$.

### 5.3. Real Data Set

Now, the proposed test will be applied to a real data set. The data set comprises the number of accidents in two different years, presented in [16], where $X$ is the accident number of the first period and $Y$ the accident number of the second period. Table 9 shows the real data set.

The $p$-value, obtained from the statistic $V_{n,w}(\hat{\theta}_n)$ of the proposed test, with $a_1 = 1$ and $a_2 = 0$ applied to the real values, is 0.838; therefore, we decided not to reject the null hypothesis, that is, the data seem to have a BHD. This is consistent with the results presented by Kemp and Papageorgiou in [26], who performed the goodness-of-fit test $\chi^2$ obtaining a $p$-value of 0.3078.

**Table 9.** Real data of *X* accident number in a period and *Y* of another period.

| | | 0 | 1 | 2 | 3 | 4 | 5 | 6 | 7 | Total |
|---|---|---|---|---|---|---|---|---|---|---|
| | | | | | | *X* | | | | |
| | 0 | 117 | 96 | 55 | 19 | 2 | 2 | 0 | 0 | 291 |
| | 1 | 61 | 69 | 47 | 27 | 8 | 5 | 1 | 0 | 218 |
| | 2 | 34 | 42 | 31 | 13 | 7 | 2 | 3 | 0 | 132 |
| *Y* | 3 | 7 | 15 | 17 | 7 | 3 | 1 | 0 | 0 | 49 |
| | 4 | 3 | 3 | 1 | 1 | 2 | 1 | 1 | 1 | 13 |
| | 5 | 2 | 1 | 0 | 0 | 0 | 0 | 0 | 0 | 3 |
| | 6 | 0 | 0 | 0 | 0 | 1 | 0 | 0 | 0 | 1 |
| | 7 | 0 | 0 | 0 | 1 | 0 | 0 | 0 | 0 | 1 |
| | Total | 224 | 226 | 150 | 68 | 23 | 11 | 5 | 1 | 708 |

**Author Contributions:** Conceptualization, F.N.-M.; methodology, F.N.-M. and P.G.-A.; software, F.N.-M. and P.G.-A.; validation, F.N.-M. and P.G.-A.; formal analysis, F.N.-M. and P.G.-A.; investigation, F.N.-M. and P.G.-A.; resources, F.N.-M.; data curation, P.G.-A.; writing—original draft preparation, F.N.-M. and P.G.-A.; writing—review and editing, F.N.-M. and P.G.-A.; visualization, F.N.-M. and P.G.-A. All authors have read and agreed to the published version of the manuscript.

**Funding:** This publication was supported by Universidad del Bío-Bío, DICREA [2220529 IF/R] and Universidad Adventista de Chile, DI [2021-139 II], Chile.

**Institutional Review Board Statement:** Not applicable.

**Informed Consent Statement:** Not applicable.

**Data Availability Statement:** Not applicable

**Acknowledgments:** The corresponding author would like to thank research project DIUBB 2220529 IF/R and Fondo de Apoyo a la Participación a Eventos Internacionales (FAPEI) at Universidad del Bío-Bío, Chile. He also thanks the anonymous reviewers and the editor of this journal for their valuable time and their careful comments and suggestions with which the quality of this paper has been improved.

**Conflicts of Interest:** The authors declare no conflict of interest.

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
