# Peer review of "Goodness-of-Fit Test for the Bivariate Hermite Distribution"

_axioms, doi:10.3390/axioms12010007_

Round 1

Reviewer 1 Report

The manuscript is written in a very structured, clear, and coherent manner. However, the content of the research raises several critical questions.

1.     The list of references consists of 17 sources, the latest from 2019. Furthermore, this is a co-authored paper of this manuscript. Therefore, I suggest adding at least 5-6 articles from 2020 and newer to the list.

2.     You are dealing with the fundamental and very broad goodness-of-fit test problem. I understand that you narrow it down to the bivariate Hermite distribution. However, the way the introduction is presented gives the reader the impression that this field of research is forgotten and neglected. It certainly is not. Since the chi-square test was proposed and analyzed by Pearson in 1900, to this day, new goodness-of-fit tests are constructed and applied to continuous and discrete data. Expand the list of literature sources and the review with articles such as

1)     Arnastauskaite, et al. A New Goodness of Fit Test for Multivariate Normality and Comparative Simulation Study. Mathematics 2021, 9, 3003. https://doi.org/10.3390/math9233003

2)     Di Noia et al. Goodness-of-fit test for count distributions with finite second moment. Journal of Nonparametric Statistics 2022, 1-19. https://doi.org/10.1080/10485252.2022.2137728

3)     Erlemann, R. and Lindqvist, B.H. 2022 "Conditional Goodness-of-Fit Tests for Discrete Distributions" Journal of Statistical Theory and Practice vol. 16, no. 8. https://doi.org/10.1007/s42519-021-00240-w

3. The simulation study is limited to the method presented in the manuscript. Moreover, it is not clear what its advantages or disadvantages are compared to the tests proposed by other researchers. Perform simulations with at least two well-known tests and expand the discussion of the results. This should help lead to the formulation of the key conclusion that concludes the paper.

Author Response

Response to Reviewer 1 Comments

Point 1: The list of references consists of 17 sources, the latest from 2019. Furthermore, this is a co-authored paper of this manuscript. Therefore, I suggest adding at least 5-6 articles from 2020 and newer to the list.

Response 1: We have added 9 new references: 3 articles from 2020, 4 articles from 2021 and 2 articles from 2022.

Point 2: You are dealing with the fundamental and very broad goodness-of-fit test problem. I understand that you narrow it down to the bivariate Hermite distribution. However, the way the introduction is presented gives the reader the impression that this field of research is forgotten and neglected. It certainly is not. Since the chi-square test was proposed and analyzed by Pearson in 1900, to this day, new goodness-of-fit tests are constructed and applied to continuous and discrete data. Expand the list of literature sources and the review with articles such as

1)     Arnastauskaite, et al. A New Goodness of Fit Test for Multivariate Normality and Comparative Simulation Study. Mathematics 2021, 9, 3003. https://doi.org/10.3390/math9233003

2)     Di Noia et al. Goodness-of-fit test for count distributions with finite second moment. Journal of Nonparametric Statistics 2022, 1-19. https://doi.org/10.1080/10485252.2022.2137728

3)     Erlemann, R. and Lindqvist, B.H. 2022 "Conditional Goodness-of-Fit Tests for Discrete Distributions" Journal of Statistical Theory and Practice vol. 16, no. 8. https://doi.org/10.1007/s42519-021-00240-w

Response 2: We have reformulated the Introduction highlighting the fact commented by the Reviewer, that from 1900 until today new goodness-of-fit tests have continued to be built and applied. Furthermore, we emphasize that our research is oriented to the discrete case and specifically to the bivariate Hermite distribution. Also, we have included the 3 references suggested by the Reviewer and we have added 6 new references.

Point 3: The simulation study is limited to the method presented in the manuscript. Moreover, it is not clear what its advantages or disadvantages are compared to the tests proposed by other researchers. Perform simulations with at least two well-known tests and expand the discussion of the results. This should help lead to the formulation of the key conclusion that concludes the paper.

Response 3: Indeed, the simulation study is limited to the test proposed in the manuscript. The reason is that we have not found another goodness-of-fit test for the bivariate Hermite distribution with which we can compare advantages, disadvantages, or simulated studies. Therefore, thanks to what was commented by the Reviewer, in this new version of the manuscript we make it explicit in Chapter 5.

Reviewer 2 Report

The manuscript seems solid. The author proposed a Cram r-von Mises-type test based on the empirical probability generation function. Then conducted a simulation study investigates the goodness of the bootstrap approach for finite sample sizes.

The English writing needs to be improved. The Introduction seems unorganized. 

The authors need to proofread before the submission.  For example, the authors write  "In the next section we will develop our statistician."

Author Response

Response to Reviewer 2 Comments

Point 1: The manuscript seems solid. The author proposed a Cramér-von Mises-type test based on the empirical probability generation function. Then conducted a simulation study investigates the goodness of the bootstrap approach for finite sample sizes.

Response 1: We appreciate the first comment made by the Reviewer.

Point 2: The English writing needs to be improved. The Introduction seems unorganized.

Response 2: Thanks to the observation made by the Reviewer, we have reformulated the Introduction, as you will be able to review it in the new version of the manuscript. In addition, we have improved the English writing.

Point 3: The authors need to proofread before the submission.  For example, the authors write  "In the next section we will develop our statistician."

Response 3: We have rephrased what was expressed by the Reviewer and we appreciate his comment.

“The following section is dedicated to developing the statistic proposed in this study and for this it is essential to know the result that is presented below, the proof of which can be reviewed in [19].”

Round 2

Reviewer 1 Report

The paper is improved according the remarks in the review and could be published in the presented form.

I accept the revised version.